# High DAPK1 Expression Promotes Tumor Metastasis of Gastric Cancer

**DOI:** 10.3390/biology11101488

**Published:** 2022-10-11

**Authors:** Qingshui Wang, Shuyun Weng, Yuqin Sun, Youyu Lin, Wenting Zhong, Hang Fai Kwok, Yao Lin

**Affiliations:** 1Central Laboratory at the Second Affiliated Hospital of Fujian Traditional Chinese Medical University, Fujian-Macao Science and Technology Cooperation Base of Traditional Chinese Medicine-Oriented Chronic Disease Prevention and Treatment, Innovation and Transformation Center, Fujian University of Traditional Chinese Medicine, Fuzhou 350001, China; 2Cancer Centre, Faculty of Health Sciences, University of Macau, Avenida de Universidade, Taipa, Macau SAR, China; 3Key Laboratory of Optoelectronic Science and Technology for Medicine of Ministry of Education, College of Life Sciences, Fujian Normal University, Fuzhou 350001, China; 4Department of General Surgery, Zhangzhou Affiliated Hospital of Fujian Medical University, Zhangzhou 363000, China; 5MoE Frontiers Science Center for Precision Oncology, University of Macau, Avenida de Universidade, Taipa, Macau SAR, China; 6Department of Biomedical Sciences, Faculty of Health Sciences, University of Macau, Avenida de Universidade, Taipa, Macau SAR, China; 7Collaborative Innovation Center for Rehabilitation Technology, Fujian University of Traditional Chinese Medicine, Fuzhou 350001, China

**Keywords:** DAPK1, gastric cancer, suppressor, prognosis, prognostic signature

## Abstract

**Simple Summary:**

Gastric cancer is a common upper gastrointestinal tumor. Death-associated protein kinase (DAPK1) was found to participate in the development of various malignant tumors. However, there are few reports on DAPK1 in gastric cancer. In this study, we found that the expression of DAPK1 was up-regulated in gastric cancer tissues. Survival analysis showed that low expression of DAPK1 was a favorable prognostic factor of overall survival and disease-free survival for gastric cancer patients. The functional experiments demonstrated that DAPK1 can promote the migration and invasion of gastric cancer cells. Meanwhile, a risk score module for four DAPK1-related genes was constructed and validated. In conclusion, these results indicated that DAPK1 can promote gastric cancer cell migration and invasion and established a signature of four DAPK1-related genes for gastric cancer that could independently predict the survival of gastric cancer patients.

**Abstract:**

Gastric cancer (GC) is a common upper gastrointestinal tumor. Death-associated protein kinase (DAPK1) was found to participate in the development of various malignant tumors. However, there are few reports on DAPK1 in gastric cancer. In this study, the TCGA and GEO datasets were used to explore the expression and role of DAPK1 in gastric cancer. The functions of DAPK1 in gastric cancer were determined by proliferation, migration and invasion assays. In addition, genes co-expressed with DAPK1 in gastric cancer were estimated through the WGCNA and correlation analysis. A DAPK1-related gene prognostic model was constructed using the Cox regression and lasso analyses. The expression of DAPK1 was significantly up-regulated in gastric cancer tissues. Kaplan–Meier analysis showed that low expression of DAPK1 was a favorable prognostic factor of overall survival and disease-free survival for gastric cancer patients. Functional experiments demonstrated that DAPK1 can promote the migration and invasion of gastric cancer cells. WGCNA, correlation analysis, Cox regression, and lasso analyses were applied to construct the DAPK1-related prognostic model. The prognostic value of this prognostic model of DAPK1-related genes was further successfully validated in an independent database. Our results indicated that DAPK1 can promote gastric cancer cell migration and invasion and established four DAPK1-related signature genes for gastric cancer that could independently predict the survival of GC patients.

## 1. Introduction

Gastric cancer (GC) is one of the highest incidences of cancer, with the fifth highest incidence rate and a global mortality rate of 9% [1]. At present, the main methods of treatment are surgery combined with chemotherapy and radiotherapy. However, there is still an unsatisfactory prognosis for people with advanced GC. GC remains an important health problem due to its high incidence. High-risk elements for GC include Helicobacter pylori infection, smoking and an unhealthy diet [2]. Moreover, oncogenesis and progression in GC are known to be affected by genetic and epigenetic alterations, such as the deactivation of tumor suppressor genes and the inactivation of oncogenes.

Death-related protein kinase (DAPK1) is a calcium-dependent serine/threonine protein kinase that was originally discovered in the regulation of apoptosis and has been confirmed to play an important role in some kinds of biological processes, including the genesis and development of tumors [3]. DAPK1 might be implicated in a variety of cell death processes in the context of various internal and external apoptotic stimuli (e.g., tumor necrosis factor-alpha) and induces and further mediates pro-apoptotic pathways [4,5]. Previous studies reported that DAPK1 was decreased in many kinds of cancer, including lung, colon, pancreatic and liver cancers [6,7,8,9]. Our previous study has indicated that DAPK1 and DDX20 (DEAD-box helicase 20) can suppress liver cancer cell invasion and migration, and the role of DAPK1 on the function of liver cancer depends on DDX20 [9]. However, the expression of DDX20 has been found to be increased in GC tissues and is an unfavorable predictor factor for GC patients. Furthermore, it is able to increase the growth and migration of GC cells [10]. Guo et al. found that DAPK1 expression was decreased in 32 GC tissues compared with 32 normal tissues based on the GSE65801 dataset, and DAPK1 promotes the killing ability of NK cells and suppresses the immune evasion of GC cells [11]. At present, there is no report on the function of DAPK1 on the proliferation, invasion and migration of GC cells.

In the study, we examined the expression and prognosis of DAPK1 in GC and investigated the effect of DAPK1 on the invasion and migration of GC cells.

## 2. Materials and Methods

### 2.1. Data Mining from Public Databases

DAPK1 expression data were obtained from TCGA (The Cancer Genome Atlas; https://portal.gdc.cancer.gov/, accessed on 14 October 2020) and GEO (Gene Expression Omnibus; http://www.ncbi.nlm.nih.gov/geo/, accessed on 18 October 2020) databases, including GSE13195, GSE63089, GSE33335 and GSE62254. The GSE13195 dataset contained 21 GC tissues and 25 normal tissues. For the remaining two datasets, 25 GC tissues and 25 normal tissues were contained in GSE33335 dataset, and 45 GC tissues and 45 normal tissues were contained in the GSE63089 dataset. The method for extracting microarray gene expression values was based on our previous study [10,12,13].

DAPK1 expression between GC tissues and normal tissues was obtained from UALCAN based on the TCGA database (http://ualcan.path.uab.edu/analysis.html, accessed on 20 October 2020) [14]. The prognostic significance of the DAPK1 mRNA for GC patients was analyzed by Kaplan–Meier using the GSE62254 database and TCGA database. UCSC XENA was used to verify the correlation of DAPK1 expression with 24 DAPK1 CpG sites (http://xena.ucsc.edu/, accessed on 28 October 2020).

### 2.2. Cells Culture and Transfection

MGC-803 cells were grown in PRMI-1640 medium (Biological Industries, Kibbutz Beit, Haemek, Israel) with 100 U/mL penicillin and 0.1 mg/mL streptomycin (BBI Life Sciences, Shanghai, China) and 10% FBS (BI, Kibbutz Beit, Haemek, Israel) at 37 °C with 5% CO_2_. The shRNA sequence of DAPK1 was cloned into the pLVX vector. The target sequence of DAPK1-specific shRNA was as follows: 5′-CAAGAAACGTTAGCAAATG-3′. Transfection was performed with lipofectamine 2000 (Invitrogen, Carlsbad, CA, USA).

### 2.3. Cell Migration Assay

MGC-803 cell migration was detected using the scratch method. Cells were cultured in six-well plates at 5.0 × 10^5^ cells/well. When cell confluence reached 100%, a cell-free zone was formed with a 10 μL pipette tip. MGC-803 cells were washed several times with PBS to remove all dead cells or debris that had scratched and floated in the culture wells. Serum-free medium was added to the plate. Cells were imaged at 0 and 48 h after scratch.

### 2.4. Cell Invasion Assay

MGC-803 cell invasion was performed with a transwell chamber. A transwell chamber with matrigel (Corning, NY, USA) was put in a 24-well plate with serum medium at 37 °C for 20 min. The 200 μL serum-free medium containing 15,000 MGC-803 cells was added to the upper part of the chamber of the transwell. After 24 h, a cotton swab was used to gently wipe off the upper basement membrane cells. Finally, the cells were fixed with 70% methanol and stained with 300 μL of Gentian violet dye solution for 10 min. After staining, excess dye in and around the basement membrane was removed with a cotton swab. Photographs of the chambers were taken.

### 2.5. CCK-8 Assay

Cells (1500 cells/per well) were cultured in 96-well plates with 6 replicates per group and incubated at 5% CO_2_ and 37 °C for 0, 24, 48 and 72 h. At each time point, all medium in the wells was aspirated off, and 10 μL CCK-8 solution (Transgen Biotech, Beijing, China) and 100 μL complete medium were added and incubated for about 2 h. The absorbance was detected at 450 nm.

### 2.6. WGCNA (Weighted Gene Co-Expression Network Analysis) Analysis

The WGCNA was performed with the WGCNA R package. Scale-free R^2^ = 0.95 and powers β = 3 were chosen as soft threshold parameters to ensure the signing of scale-free co-expression gene networks. A cluster dendrogram was created based on the topological overlap matrix with a minimum cluster size of 20. In total, 23 modules were generated.

### 2.7. Risk Score Module Construction

The GSE62254 transcriptome data was used to build the risk score module of DAPK1 related gene using Lasso (least absolute shrinkage and selection operator). The L1-norm was applied to penalize the weights of the features. The risk score module based on DAPK1-related gene characteristics was built based on their weight calculated by lasso analysis. Risk score = (0.788 ∗ FAM161B (FAM161 Centrosomal Protein B) + 0.741 ∗ SEC31A (SEC31 Homolog A, COPII Coat Complex Component) + 1.568 ∗ YPEL5 + 0.197 ∗ ZSWIM6 (Zinc Finger SWIM-Type Containing 6)).

### 2.8. Functional Enrichment Analysis

To analyze the function of DAPK1-related genes in GC tissues, the GO-BP (Gene Ontology-Biological Process) and KEGG (Kyoto Encyclopedia of Genes and Genomes) pathways were analyzed through the DAVID database (http://david.abcc.ncifcrf.gov, accessed on 29 October 2020).

### 2.9. Correlogram Analyses

The correlation plots were drawn by use of the program R package. Correlations were considered as significant when *p* < 0.05 and |*r*| > 0.3. Significant correlations are shown as red or blue points on the correlation plot. The magnitudes of the points in the correlation matrices are proportional to the strength of the correlation.

### 2.10. Statistical Analysis

Student’s *t*-test was used to generate a *p*-value for comparison between normal and GC tissues. DFS (disease-free survival) and OS (overall survival) were calculated through Kaplan–Meier, and differences between groups were tested by the log-rank test. Here, *p* < 0.05 was considered statistically significant.

## 3. Results

### 3.1. The Expression Pattern and Prognosis Analysis of DAPK1 in GC Patients

In this research, three related GEO datasets, including GSE13195, GSE63089 [15] and GSE33335 [16], were assessed to explore the expression patterns of DAPK1 in GC. The above three GEO datasets were based on the Affymetrix Human Exon 1.0 ST Array. We found that DAPK1 was significantly up-regulated in GC tissues based on GSE13195 (Figure 1A), GSE63089 (Figure 1B), and GSE33335 (Figure 1C). To minimize the potential batch effects for three independent databases, batch effect removal strategies were performed by the COMBAT algorithm. GSE13195, GSE63089, and GSE33335 were merged to remove batch effects and renamed as 3GEOs. From the boxplot, we can observe that the sample distribution of each dataset before removing the batch effect is different, suggesting that there is a batch effect. After removing batch effects, the data distributions between each dataset tend to be consistent (Figure 1D). We found that DAPK1 expression was increased in GC tissues based on the 3GEO database (Figure 1E). Based on the TCGA dataset, we also observed that DAPK1 expression was increased in GC tissues (Figure 1F).

Because the GSE62254 dataset has a sufficient sample size (*n* = 300) and complete recurrence information, we chose the GSE62254 dataset to perform Kaplan–Meier analysis for the OS and DFS. According to the prognosis analysis, low DAPK1 levels were correlated with better OS and DFS (Figure 2A,B). Based on TCGA dataset, low DAPK1 levels were also linked to greater OS in GC patients (Figure 2C).

Table 1 summarises the clinicopathological characteristics of GC patients in the GSE62254 dataset. DAPK1 expression was significantly correlated with the sex and GC stage. 

Meanwhile, we performed univariate and multivariate Cox analyses to ascertain the independent prognostic factor for GC patients (Table 2 and Table 3). Tumor stage, pathology and DAPK1 were identified as independent prognostic factors by univariate Cox analysis. Through multivariate analysis, the tumor stage was established as an independent prognostic factor of GC patients.

### 3.2. The Relationships of DAPK1 Expression with DNA Methylation and Transcription Factors 

DAPK hypermethylation occurs in many different cancers than in normal tissues. Thus, we detected associations between DAPK1 expression and DNA methylation via the UCSC XENA website. The methylation values of 24 methylation sites in the promoter of DAPK1 are shown in Figure 3A. The correlation of 24 DAPK1 CpG sites with DAPK1 expression was analyzed. Results indicated that DAPK1 expression is inversely correlated with 5 CpG sites, including cg19734228, cg15746719, cg14014720, cg13527872 and cg24754277 (Figure 3B–F).

To ascertain membership of the putative molecular network which regulates DAPK1 expression, we investigated transcription factors potentially affecting DAPK1 gene transcription. First, the top 20 regulatory transcription factors in human cancers were identified using the Cistrome DB Toolkit (Figure 3G). Subsequent analyses revealed no correlation between DAPK1 and the expression of 20 transcription factors (Figure 3H).

### 3.3. DAPK1 Displays Oncogenic Features in GC Cells

Prompted by the above findings, we examined whether DAPK1 exerted oncogenic functions in GC. Firstly, DAPK1 was knocked down in MGC-803 cells using the DAPK1-shRNA plasmid (Figure 4A). Knockdown of endogenous DAPK1 did not affect MGC-803 cell proliferation (Figure 4B). Migration assay and invasion assay demonstrated that DAPK1 knockdown suppressed the invasion and migration abilities of MGC-803 cells (Figure 4C,D).

DAPK1 overexpression plasmid has been further transfected into MGC-803 cells to analyze the effects of DAPK1 on GC cells (Figure 5A). In Figure 5B, overexpression of DAPK1 did not affect MGC-803 cell proliferation. The migration and invasion assays showed that increased DAPK1 levels promoted the invasion and migration abilities of MGC-803 cells (Figure 5C,D).

### 3.4. Construction of a Co-Expression Network for Weighted Genes of DAPK1 in GC

WGCNA was used to explore networks of the co-expression of the DAPK1 gene in GC patients. We constructed WGCNA using the GSE62254 dataset and calculated network topologies with soft threshold power values to select the optimal threshold. A power value of 3 is the minimum power for scale-free topology in GSE62254 databases (Figure 6A,B). After that, the co-expression resemblance matrix was transformed into an adjacency matrix by choosing 3 as a soft threshold (Figure 6C). We used the dynamic tree-cutting method and yielded 23 meaningful modules. A total of 1367 genes co-expressed with DAPK1 lengthened to the blue module. KEGG-pathway and GO-BP analysis were used to reveal the functions of these 1367 genes. The result of GO-BP demonstrated that the top 10 terms were cell cycle, cell cycle process, etc. (Figure 6D). KEGG-pathway results demonstrated that the top 10 terms were cell cycle, DNA replication, etc. (Figure 6E).

### 3.5. Prognosis Model of DAPK1-Related Genes Constructed

Correlation analysis between the expression of 1367 genes and DAPK1 expression was performed, and it was found that only the expression of 12 genes was positively correlated with DAPK1 expression in GC, including FAM161B (family with sequence similarity 161, member B), PELI2 (pellino E3 ubiquitin-protein ligase family member 2), FNIP2 (folliculin interacting protein 2), GRAMD1B (GRAM domain containing 1B), IFT20 (intraflagellar transport 20), NTN4 (netrin 4), PDGFD (platelet-derived growth factor D) SEC31A (SEC31 homolog A), STX12 (syntaxin 12), UGCG (UDP-glucose ceramide glucosyltransferase), YPEL5 (yippee-like 5) and ZSWIM6 (zinc finger, SWIM-type containing 6) (r > 0.3, *p* < 0.05) (Figure 7A). The survival analysis results indicated that high expression of YPEL5, FAM161B, SEC31A, ZSWIM6, and PDGFD was linked with poor survival of GC patients (*p* < 0.05) (Figure 7B).

Next, these 5 genes were included in the LASSO analysis. Figure 7C plotted the partial likelihood deviance versus log (λ), where λ was the tuning parameter. The value of λ was 0.03 and was chosen by 10-fold cross-validation. Figure 7C shows the change in the trajectory of each prognostic variable. Each variable has a coefficient as its weight provided by LASSO. We obtained 4 variables with nonzero coefficients at the value λ chosen by the cross-validation. These prognostic variables included FAM161B, SEC31A, YPEL5, and ZSWIM6. The variables were ranked by their contribution to the prognostic model as follows: FAM161B (weight: 0.788), SEC31A (weight: 0.741), YPEL5 (weight: 1.568) and ZSWIM6 (weight: 0.197). Then, the prognostic model risk score for each patient was computed according to the summation of 4 variables multiplied by a coefficient generated from the LASSO regression. The formula for this risk score was (0.788 * FAM161B expression) + (0.741 * SEC31A expression) + (1.568 * YPEL5 expression) + (0.197 * ZSWIM6 expression). Each GC patient was classified into low- and high-risk score groups based on the median value. The distribution of survival status, risk scores, and the expression of four genes of GC patients in the GSE62254 dataset are shown in Figure 7E. The survival analysis results indicated that the prognosis of GC patients in the high-risk group was inferior to the low-risk group (Figure 7F).

Moreover, survival analyses were performed on GC patients with low- and high-risk scores in different subgroups, including age, gender and tumor stage (Figure 8). These results reinforce the powerful stratification capabilities of the risk scores.

To demonstrate the reliability of the risk score model, the TCGA database was used as the verification dataset (Figure 9). GC patients were divided into the low-and high-risk group, respectively. GC patients with high-risk scores had a worse OS compared to the low-risk score group, indicating that the model has excellent accuracy.

### 3.6. Construction of a Clinical Prognostic Prediction Model

A survival nomogram has been generated to accurately compute the 1-, 3- and 5-year survival probabilities based on gender, age, tumor stage and risk score (Figure 10A). As shown in Figure 10B, the nomogram performed well compared to the ideal model. We consider that the nomogram has been shown to have excellent accuracy in predicting survival in GC patients.

## 4. Discussion

DAPK1 has an important role in the regulation of cellular processes and is involved in tumorigenesis and development [17,18,19,20]. DAPK1 is emerging as a tumor suppressor for some cancers, including bladder, liver, colorectal and oesophageal squamous cell carcinomas [9,21,22,23]. In liver cancer, DAPK1 expression is markedly down-regulated; low expression of DAPK1 is an unfavorable prognosis for liver cancer patients. DAPK1 can inhibit the invasion and migration of liver cancer cells by up-regulating DDX20 [9]. In GC tissues, DDX20 is significantly up-regulated, and high expression of DDX20 is an unfavorable prognostic factor for GC patients. DDX20 can promote the proliferation, invasion and migration of GC cells [10]. Therefore, DDX20, the downstream gene of DAPK1, appears to be both an oncogene and an oncogene suppressor in different tumor types. Recent studies have demonstrated the oncogenic role of DAPK1 [24]. We were, therefore, interested in the function of DAPK1 in GC cells.

A previous study reported that DAPK1 was down-regulated in 32 GC tissues compared with 32 normal tissues based on the GSE65801 dataset [11]. Because the sample size of GSE65801 is relatively small, we used a larger sample size to detect the expression and prognosis of DAPK1 in GC based on three GEO and TCGA datasets. In this study, three GEO datasets (GSE13195, GSE63089 and GSE33335) and the TCGA dataset demonstrated a significant up-regulation of DAPK1 expression in GC tissues. Survival analysis suggested that the low expression of DAPK1 was a favorable prognostic factor for GC patients. These results demonstrated that DAPK1 may be a carcinogenic gene in GC. Guo et al. found that DAPK1 was associated with the immune microenvironment of GC. DAPK1 is able to increase the killing ability of NK cells and restrain immune evasion of GC cells [11]. However, the effects of DAPK1 on the proliferation, invasion and migration of GC cells have not been reported. Therefore, we evaluated the effect of DAPK1 on the proliferation, invasion and migration of GC cells. Results have shown that DAPK1 can promote the invasion and migration of GC cells. These results indicate that DAPK1 can significantly influence invasion and immune escape in GC and suggest that DAPK1 plays different biological roles both at the tissue level and at the cellular level. Furthermore, further single-cell analysis studies will be necessary to improve the comprehension of the differences in DAPK1 expression in GC.

Compared to normal tissue, hypermethylation of DAPK1 has been routinely observed in many cancers, including lung, kidney and bladder cancers [25,26,27]. In the study, we observed that DAPK1 expression was inversely associated with cg19734228, cg15746719, cg14014720, cg13527872 and cg24754277 based on the TCGA database. Thus, we have suggested that DAPK1 expression is modulated by DNA methylation in GC.

Next, we used the WGCNA technique to identify DAPK1-related gene modules in GC and identified the blue module as the DAPK1-related gene module. Functional analyses found cell cycle and DNA metabolic process pathways were enriched in the DAPK1-related gene module. Correlation analysis and lasso analysis were employed to build a DAPK1-related risk score model consisting of four DAPK1-related genes, including FAM161B, SEC31A, YPEL5 and ZSWIM6, which could classify patients into low-risk and high-risk groups. There have been few reports on the roles of FAM161B in tumors. Previous studies have reported that FAM161B was regulated by methylation in colon cancer [28]. SEC31A is an external component of the COPII associated with the pathogenesis of non-small cell lung cancer [29]. YPEL5 is a member of the highly conserved YPEL gene family [30]. YPEL5 has been found to inhibit the proliferation of cervical and colorectal cancer cells [31,32]. The expression of ZSWIM6 was found to be increased in the brain. Previous studies found that mutations of the ZSWIM6 gene affect developmental and neurological disorders [33,34,35]. Subsequently, the model was validated in the TCGA validation database. We refined the model by developing a nomogram to make it more clinically applicable.

## 5. Conclusions

In conclusion, we identified that DAPK1 is highly expressed in GC and is associated with a poor prognosis. Aside from this, we proved that DAPK1 increased the invasion and migration of GC cells. A DAPK1-related multi-gene signature prognostic model, including co-expression genes, was constructed, which might have crucial implications in predicting the prognosis of GC patients

## Figures and Tables

**Figure 1 biology-11-01488-f001:**
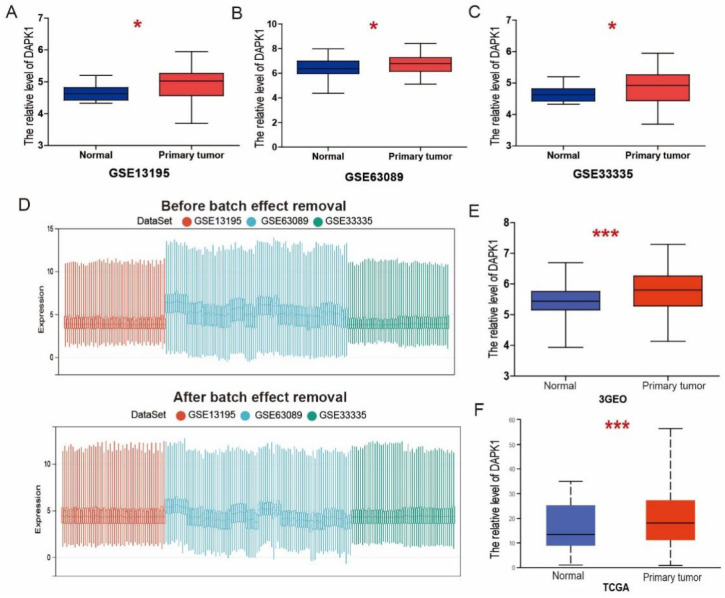
DAPK1 expression in GC. (**A**–**C**) The expression of DAPK1 in GC based on GSE13195 (**A**), GSE63089 (**B**), and GSE33335 (**C**). (**D**) Boxplots of data distribution before and after batch effect removal for GSE13195, GSE63089, and GSE33335 datasets. (**E**,**F**) The expression of DAPK1 in GC based on 3GEOs (**E**) and TCGA (**F**). * *p* < 0.05; *** *p* < 0.001.

**Figure 2 biology-11-01488-f002:**
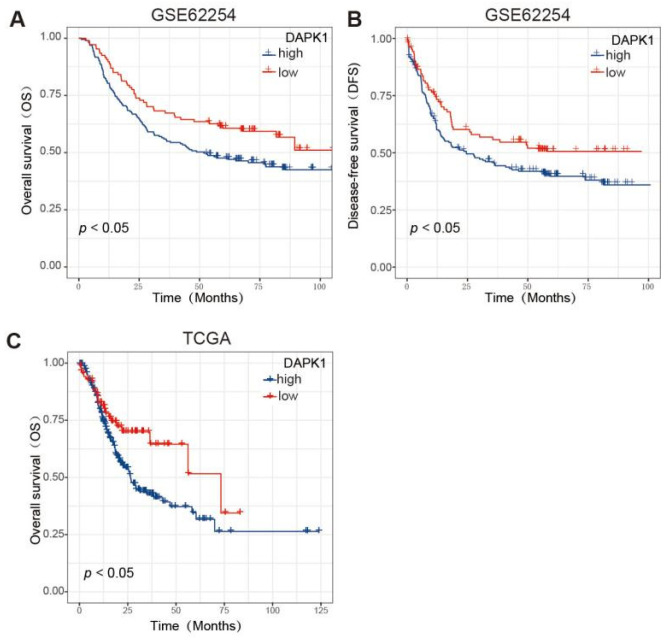
The prognosis of DAPK1 in GC patients. (**A**) Survival analysis of DAPK1 on the OS of GC patients based on the GSE62254 dataset; (**B**) survival analysis of DAPK1 on the DFS of GC patients based on the GSE62254 dataset; (**C**) survival analysis of DAPK1 on the OS of GC patients based on the TCGA dataset.

**Figure 3 biology-11-01488-f003:**
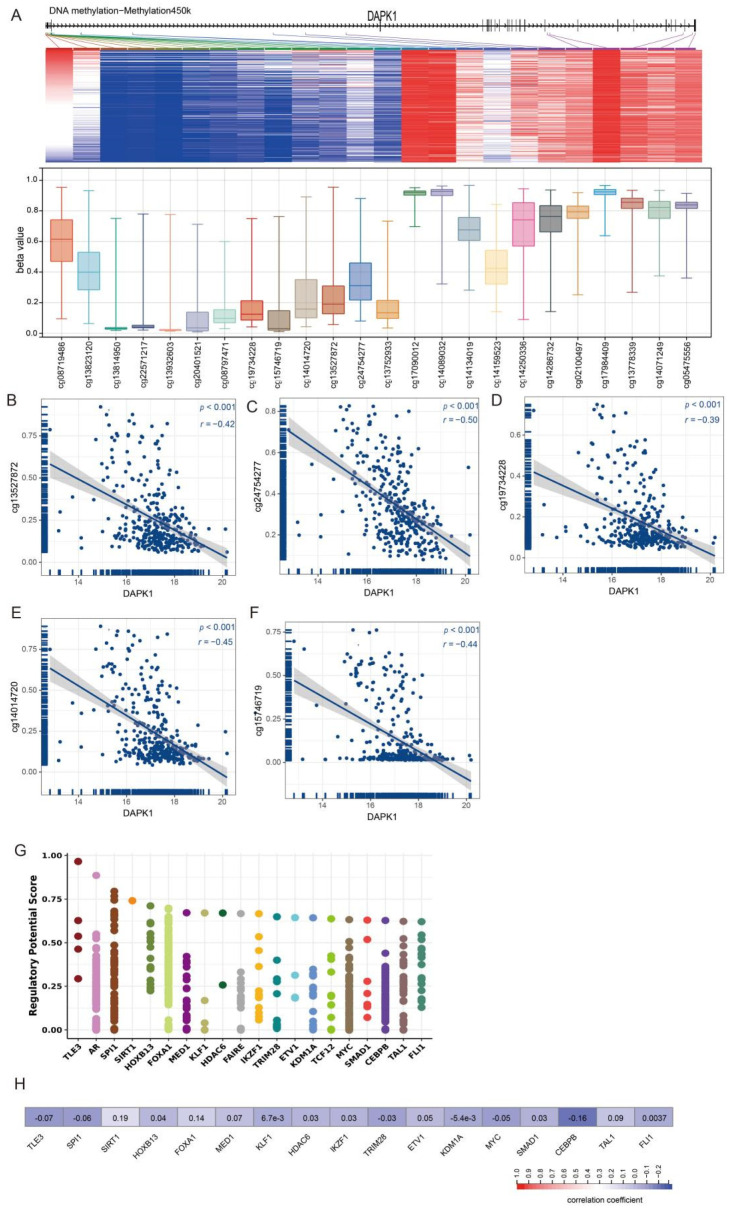
The correlation between the level of DAPK1 and methylation of DAPK1 DNA CpG sites and TF. (**A**)The expression value of 24 methylation sites of DAPK1 promoter. (**B**–**F**) The correlation of DAPK1 with (**B**) cg13527872, (**C**) cg2474277, (**D**) cg19734228, (**E**) cg14014720 and (**F**) cg15746719. (**G**) Top 20 TFs that potentially regulate DAPK1 in different human cancers. (**H**) Correlation between DAPK1 and TF mRNA expression.

**Figure 4 biology-11-01488-f004:**
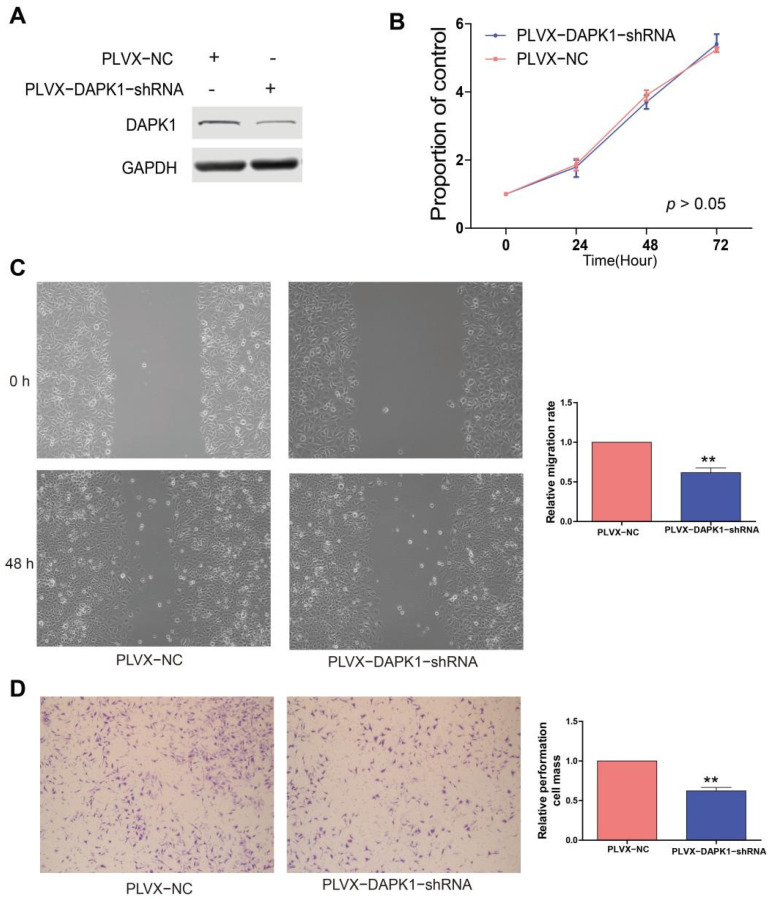
Knockdown DAPK1 inhibits GC cell migration and invasion. (**A**) Western blot was used to detect the levels of DAPK1 in MGC-803 cells transduced by DAPK1-shRNA; (**B**) DAPK1 knockdown did not significantly affect MGC-803 cell proliferation by using CCK8; (**C**) DAPK1 knockdown inhibited MGC-803 cell migration; (**D**) DAPK1 knockdown inhibited MGC-803 cell invasion. ** *p* < 0.01. Please find the original western blot of (**A**) in the Appendix A.

**Figure 5 biology-11-01488-f005:**
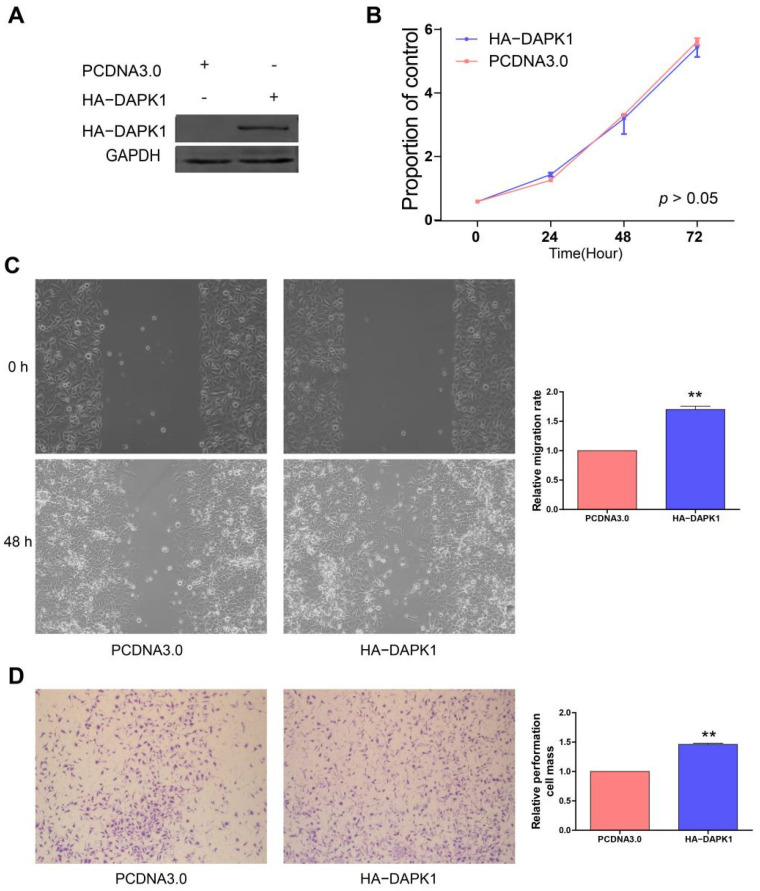
Overexpression of DAPK1 promotes GC cell migration and invasion. (**A**) Western blot was used to detect the levels of DAPK1 in MGC-803 cells transduced by HA-DAPK1; (**B**) DAPK1 did not significantly affect MGC-803 cell proliferation by using CCK8; (**C**) DAPK1 promoted MGC-803 cell migration; (**D**) DAPK1 promoted MGC-803 cell invasion. ** *p* < 0.01. Please find the original western blot of (**A**) in the Appendix A.

**Figure 6 biology-11-01488-f006:**
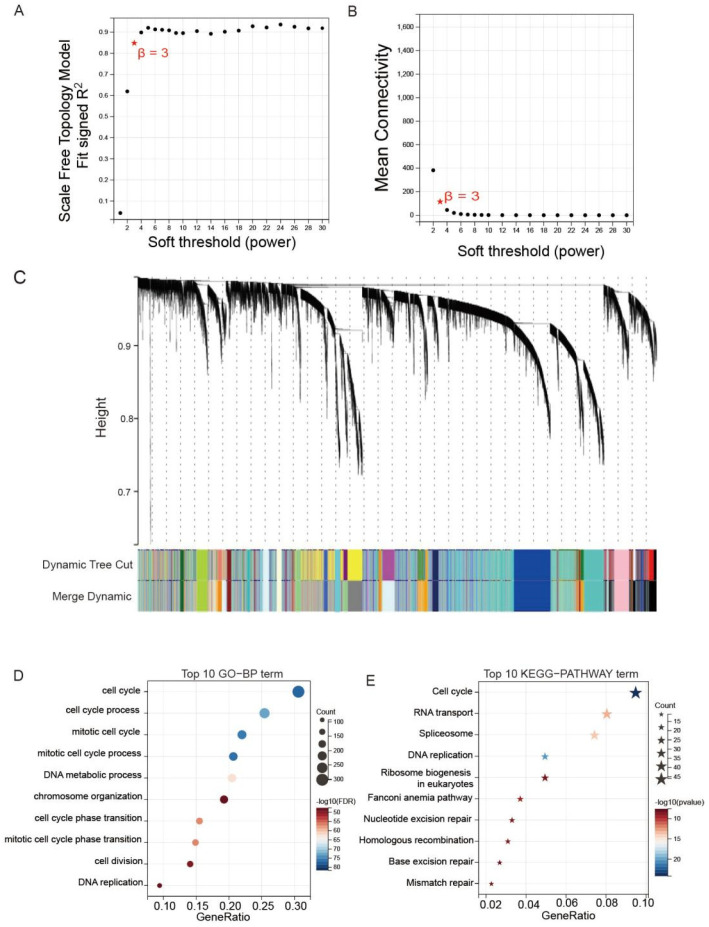
WGCNA was used to identify co-expression module genes relating to DAPK1. (**A**) Association between power and scale-free topology model fit in GSE62254 database. (**B**) Association between power and mean connectivity in GSE62254 dataset. (**C**) Dendrogram of modules identified by WGCNA in the GSE62254 dataset. (**D**) GO-BP analysis of the genes in the blue module. (**E**) KEGG-pathway analysis of the genes in the blue module.

**Figure 7 biology-11-01488-f007:**
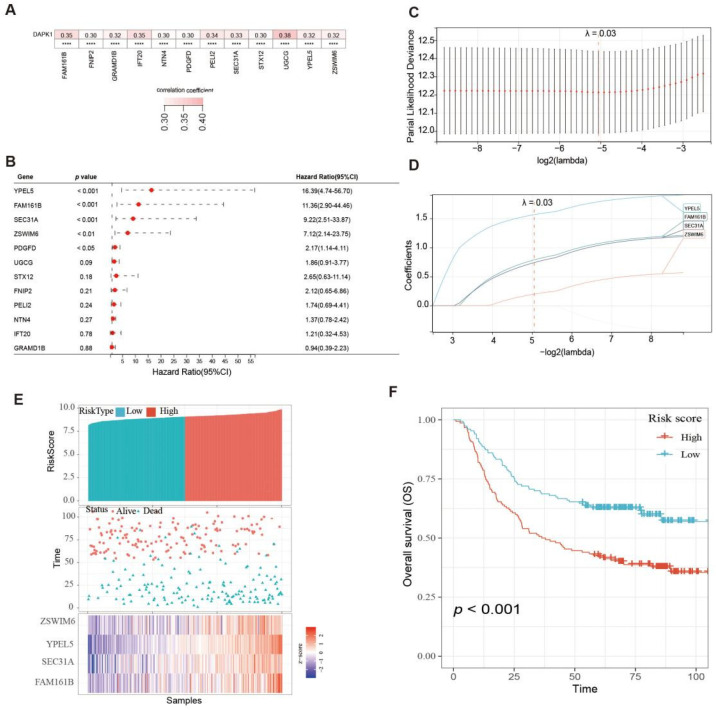
Construction of survival risk score system based on DAPK1-related genes. (**A**) The correlation between DAPK1 and DAPK1-related genes. (**B**) Survival analysis of the DAPK1-related genes. (**C**) Partial likelihood deviance of overall survival for LASSO coefficient profiles. (**D**) LASSO coefficient profiles of 4 genes for overall survival. (**E**) The distribution of survival status, risk scores, and the expression of four genes of GC patients. (**F**) Survival analysis of risk score module.

**Figure 8 biology-11-01488-f008:**
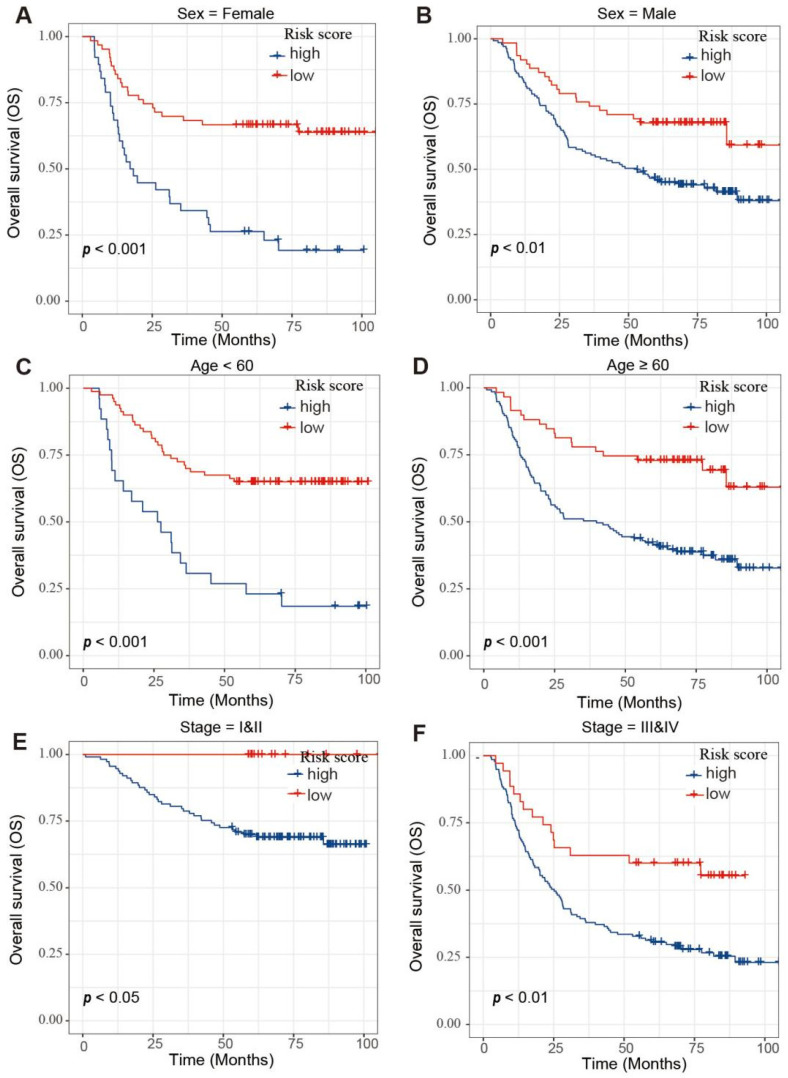
Survival analysis of GC patients in different subgroups. (**A**,**B**) Survival analysis of the GC patients with (**A**) sex = female and (**B**) GC patients with sex = male subgroup. (**C**,**D**) Survival analysis of the GC patients with age < 60 (**C**) and GC patients with age ≥ 60 (**D**) subgroup. (**E**,**F**) Survival analysis of the GC patients with stage = I & II (**E**) and GC patients with stage = III & IV (**F**) subgroup.

**Figure 9 biology-11-01488-f009:**
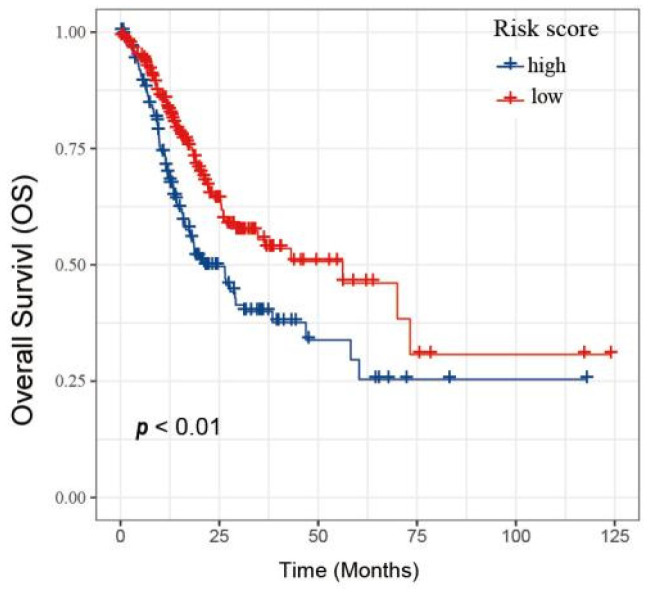
Validation of the risk score model based on TCGA database.

**Figure 10 biology-11-01488-f010:**
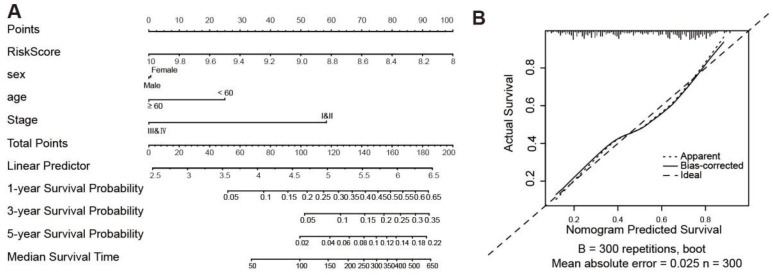
Nomogram for the prediction of outcomes of GC patients. (**A**) Nomogram for predicting survival that combines clinical data. (**B**) The calibration plots for predicting OS.

**Table 1 biology-11-01488-t001:** Characteristics of GC patients and DAPK1 level based on GSE62254 dataset.

Characteristics	All Patients		Low DAPK1	High DAPK1	*p* Value
	*n* = 300		*n* = 107	*n* = 193	
	No.	(%)	No.	No.	
Age					0.1168
≤50	41	13.7	10	31	
>50	259	86.3	97	162	
Sex					0.0110
Male	199	66.3	81	118	
Female	101	33.7	26	75	
Tumor Stage					0.0105
I + II	127	42.3	56	71	
III + IV	173	57.7	51	122	
Pathology					0.2796
Diffuse	135	45	42	93	
Intestinal	146	48.7	58	88	
Mixed	17	5.6	5	12	

**Table 2 biology-11-01488-t002:** A univariate Cox analysis for OS and DFS of GC patients based on the GSE62254 dataset.

Variables	OS		DFS	
	HR (95% CI)	*p*	HR (95% CI)	*p*
Age (years)				
≤50	Reference		Reference	
>50	1.24 (0.8–1.92)	0.330	1.19 (0.78–1.83)	0.420
Sex				
Female	Reference		Reference	
Male	1.11(0.79–1.55)	0.560	1.02(0.74–1.42)	0.900
Tumor Stage				
I + II	Reference		Reference	
III + IV	3.47 (2.39–5.05)	<0.001	3.46 (2.39–5.01)	<0.001
Pathology				
Diffuse	Reference		Reference	
Intestinal	0.6 (0.43–0.83)	0.002	0.68 (0.49–0.94)	0.019
DAPK1 expression				
Low	Reference		Reference	
High	1.49 (1.05–2.11)	0.025	1.41 (1.00–1.99)	0.046

**Table 3 biology-11-01488-t003:** A multivariate Cox analysis for OS and DFS of GC patients based on the GSE62254 dataset.

Variables	OS		DFS	
	HR (95% CI)	*p*	HR (95% CI)	*p*
Age (years)				
≤50	NA		NA	
>50				
Sex				
Female	NA		NA	
Male				
Tumor Stage				
I + II	Reference		Reference	
III + IV	3.52 (2.33–5.31)	<0.001	3.59 (2.40–5.38)	<0.001
Pathology				
diffuse	Reference		Reference	
intestinal	0.79 (0.56–1.12)	0.182	0.88 (0.62–1.25)	0.471
DAPK1 expression				
Low	Reference		Reference	
High	1.25 (0.87–1.81)	0.23	1.29 (0.89–1.86)	0.179

## Data Availability

Not applicable.

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
