# Peer review of "High DAPK1 Expression Promotes Tumor Metastasis of Gastric Cancer"

_biology, 2022, doi:10.3390/biology11101488_

Round 1
Reviewer 1 Report
Paper by Wang and co-workers have un interesting aim. In this study authors demonstrated Hight expression of DAPK1 and its association with poor prognosis. However, some minor revisions are required before publication
minor
1-delete line 212 reported the same text of line 213
2-there are typos
3-Figure 7 is difficult to read
Reviewer 2 Report
High DAPK1 expression promotes tumor metastasis of gastric 2 cancer
Overall the study gives an insight in to the role of DAPK1 role in prognosis of gastric cancers. This study demonstrates the importance of the DAPK1 in gastric tumor progression using various tools and experiments. Also, the present study demonstrates the importance of the DAPK1 levels and its positive correlation with overall survival and disease free survival rate of gastric cancer patients. Interestingly DAPK1 knockout specific to gastric cancer cell line in this study leads to migration and invasion of these cells. Taken together, this study provides evidence for a major newfound physiological role for the DAPK1 in the regulation of tumor progression of gastric cancer.
Several issues remain to be addressed with this manuscript.
Author’s needs to address the difference between their study and the study conducted by the Guo et al more precisely in the introduction and as well as in the discussion section. As, this study demonstrates completely different role of DAPK1 in gastric cancer progression were they have demonstrated low levels of DAPK1 is more beneficial for gastric cancer progression by using in-vivo and in-vitro model system. But this article suggests the opposite of this.
Did the authors observe a similar any correlation with DAPK1 high expression with the classical angiogenic markers like VEGF, CD31 in their study?
Also, why did the authors not use the GSE65801 data set in their study which is used by Guo et al?
What was result DAPK1 comparison individual data sets (GSE13195, GSE63089, and GSE33335) used in the batch analysis study? Did the authors find a similar trend in the DAPK1 expression in these data sets? Also what would be the result if they used the GSE65801 data set in their batch analysis as this data provides different insights in the DAPK1 expression pattern?
Authors need to clearly mention in the introduction about the GSE62254 data importance and why did they chose this data set to perform Kaplan-Meier analysis for the overall survival and disease free survival rate.
In the Martials & methods RPMI is written as PRMI correct this.
Also in cell invasion assay matrigel was put in a 24 well plate with serum medium and cultured at 5% Co2 and 37C for 20 min what does this sentence mean- Explain this sentence?
Materials and Methods need major revision need to very clear and need to explain the acronyms.
Authors need to provide the sequence of shRNA used in this study, did they buy it or design the shRNA is also not mentioned in the Materials and Methods.
Authors also need to provide western blot images for the DAPK1 knock out and over expression to confirm the shRNA.
Did the authors check for the classical angiogenic markers expression in the invasion experiment under knock out and over expression conditions? Also, did the authors do any RNA seq of these cells under these conditions?
Also reference’s need to be updated with latest references
Reviewer 3 Report
Comments
The study entitled “High DAPK1 expression promotes tumor metastasis of gastric cancers” is very interesting. However, while writing the paper the authors have shown some negligence that I believe can be well addressed before accepting for publication. I have the following concerns:
1. Based on our previous studies, the use of microarray gene 83 expression values [10, 12, 13]. LINE 84. Here references 12 & 13 neither talks about DAPK1 expression no are gastric cancer. Their inclusion needs further explanation.
2. The list for expanded form of different acronyms used in the paper has not been provided. It needs to be added for all acronyms for better understanding e.g.
“In this study, TCGA and GEO datasets were used to ….” Line 33
“……the WGCNA and correlation anal….” Line 36
“DFS and OS were calculated through Kaplan-Meier and differences…” line 142
3. Grammer check is needed especially in introduction. Grammatical errors examples
“In addition, Genes co-…” line 35. In the gene “g” should be in lowercase.
“WGCNA, Correlation …” line 41. C must be in lowercase.
Check and revise these sentences
“ Death-related protein kinases (DAPK1) is a calcium-dependent serine/threonine kinases line that were originally discovered in the regulation of apoptosis, and have been con-…” line 59 & 60
“Add serum-free medium and 15,000 cells in a small chamber. after 24 h, wipe off the upper basement membrane cells, fix with 70% methanol and stain with 300 μl of gentian violet dye solution for 10 min. After staining, ..” lines 108- 110
“Clustered dendrograms were created based on the topological overlap matrix with a minimum cluster size of 20, and 23 modules were built. Line 122-123
“significant correlations with p ≤ 0.05 was considered significant……: line 135-136
4. “…a DAPK1-related four-genes signature for gastric cancer..” line 45. It may be well written as four signature genes…
5. At many places full stop has been put before as well as after citation. Check all through the manuscript and remove the full stop before citation and use uniform format all through the manuscript examples
“…..global mortality rate of 9%. [1]. At present, the main methods of treat- …” line 51.
“……..smoking and an unhealthy diet. [2]. Moreover, the …” line 55
“……..further mediates pro-apoptotic pathways. [4, 5]. Previous …” line 64
“….pancreatic and liver cancers. [6-9]. Our previous study has indicated that DAPK1…” line 66
“… of GC cells. [10]. Guo et al found that DAPK1 promotes the killing ability of NK..” line 71
“…gastric cancer cells. [11]. At present, there is…” line 72
“….cg24754277. (Figure 3B-F). line 199
“….process, etc. (Figure 6D). line 248
“…..tion, etc. (Figure 6E). line 250
“..bladder cancers. [25-27]. In the study, …”LINE 342
“….cell lung cancer. [29]. YPEL5 is a member of the highly conserved YPEL gene…” LINE 355
6. Acronym for gastric cancer as GC has been used in first instance along with full form and at many other later places only GC is used. However, there is a range of instances all through the manuscript where instead of GC expanded form is repeatedly used. I would like to suggest uniform patter to be followed all through the manuscript. Examples
“….with advanced gastric cancer. Gastric cancer re-…” line 53
“,…progression in gastric cancer are known to be affected by genetic and …” line 56
“..immune evasion of gastric cancer cells. [11]. At present, there is ..” line 72
“….. DAPK1 gene in gastric cancer patients. We constructed WGCNA by using the GSE62254 dataset and calcu- …”Lines 239 240
“…gastric cancer patient had been..” line 273
“…expression of gastric cancer patients…” line 275
“…the prognosis of gastric cancer…” line 276
“….expression of gastric cancer patients. (F) Survival analysis of risk score module”. Line 283
“…on gastric cancer patients with low- and..” line 285
“….gastric cancer patients with (A) sex=female and (B) gastric cancer patients with sex=male subgroup”. line 290
“…Survival analysis of the gastric cancer patients with age… and gastric cancer patients..” line 291
“…gastric cancer patients with stage=â…¢& â…£ (F) subgroup”. Line 293
“…Gastric cancer patients with high-risk scores had a worse OS com-..” line 297
“…microenvironment of gastric cancer. DAPK1 is….” LINE 334
“… immune evasion of gastric cancer..” LINE 335
“…escape in gastric cancer and suggest that DAPK1 plays different biological roles..” LINE 337
“..gene modules in gastric cancer…” LINE 346,
“…expressed in gastric cancer and is as..” LINE 363
· “Survival analysis of DAPK1 on the OS ofGC patients based on …” line 172. There must be space between of & GC.
· “Figure 3A-F” poor quality. Replace the images with better quality.
· 3.3. DAPK1 displays oncogenic features in gastric cancer cells. Line 212. Duplicate & hence remove it.
· In (Figure 4B) & (Figure 5B), Is it proportion or propotion ? p must be capitalized.
· “Atotal of 1367” there must be space between A and total. Line 245.
· Figure 6. Poor quality, less visibility. Replace with better quality pictures.
· Figure 7. Poor quality, less visibility. Replace with better quality pictures.
· Figure 8. Poor quality, less visibility. Replace with better quality pictures.
· Figure 10. Poor quality, less visibility. Replace with better quality pictures.
· Wan, Y.; Wu, X.Y. [Expression and clinical significance of DAPK1 and CD147 in esophageal squamous cell carcinoma]. Zhonghua zhong liu za zhi [Chinese journal of oncology] 2012, 34, 44-48. Line 434-435. Remove parenthesis
· In these reference journal name is not provided in full form
· 2.Ang, T.L.; Fock, K.M. Clinical epidemiology of gastric cancer. Singapore Med J 2014, 55, 621-628, doi:10.11622/smedj.2014174.
· 9.Huang, Y.; Wang, C.; Li, K.; Ye, Y.; Shen, A.; Guo, L.; Chen, P.; Meng, C.; Wang, Q.; Yang, X.; et al. Death-associated protein kinase 1 suppresses hepatocellular carcinoma cell migration and invasion by upregulation of DEAD-box helicase 20. Cancer Sci 2020, 111, 402 2803-2813, doi:10.1111/cas.14499
· 12.Wang, Q.S.; Li, F.; Liao, Z.Q.; Li, K.; Yang, X.L.; Lin, Y.Y.; Zhao, Y.L.; Weng, S.Y.; Xia, Y.; Ye, Y.; et al. Low level of Cyclin-D1 410 correlates with worse prognosis of clear cell renal cell carcinoma patients. Cancer Med 2019, 8, 4100-4109, doi:10.1002/cam4.2313.
· 13.Niu, H.; Li, F.; Wang, Q.; Ye, Z.; Chen, Q.; Lin, Y. High expression level of MMP9 is associated with poor prognosis in patients 412 with clear cell renal carcinoma. PeerJ 2018, 6, e5050, doi:10.7717/peerj.5050.
· 22.Steinmann, S.; Kunze, P.; Hampel, C.; Eckstein, M.; Bertram Bramsen, J.; Muenzner, J.K.; Carlé, B.; Ndreshkjana, B.; Kemenes, S.; 431 Gasparini, P.; et al. DAPK1 loss triggers tumor invasion in colorectal tumor cells. Cell Death Dis 2019, 10, 895,
· 29.Cheng, F.; Yu, J.; Zhang, X.; Dai, Z.; Fang, A. CircSEC31A Promotes the Malignant Progression of Non-Small Cell Lung Cancer 448 Through Regulating SEC31A Expression via Sponging miR-376a. Cancer Manag Res 2020, 12, 11527-11539, doi:10.2147/cmar.s280124.
· 31.Barrett, T.; Wilhite, S.E.; Ledoux, P.; Evangelista, C.; Kim, I.F.; Tomashevsky, M.; Marshall, K.A.; Phillippy, K.H.; Sherman, P.M.; 453 Holko, M.; et al. NCBI GEO: archive for functional genomics data sets--update. Nucleic Acids Res 2013, 41, D991-995,
